# Determinants of Lipid Parameters in Patients without Diagnosed Cardiovascular Disease—Results of the Polish Arm of the EUROASPIRE V Survey

**DOI:** 10.3390/jcm12072738

**Published:** 2023-04-06

**Authors:** Jakub Ratajczak, Aldona Kubica, Piotr Michalski, Łukasz Pietrzykowski, Aleksandra Białczyk, Agata Kosobucka-Ozdoba, Katarzyna Bergmann, Krzysztof Buczkowski, Magdalena Krintus, Piotr Jankowski, Jacek Kubica

**Affiliations:** 1Department of Cardiac Rehabilitation and Health Promotion, Nicolaus Copernicus University in Torun, Collegium Medicum in Bydgoszcz, 85-094 Bydgoszcz, Poland; 2Department of Cardiology and Internal Medicine, Nicolaus Copernicus University in Torun, Collegium Medicum in Bydgoszcz, 85-094 Bydgoszcz, Poland; 3Students’ Scientific Circle of Cardiology, Nicolaus Copernicus University in Torun, Collegium Medicum in Bydgoszcz, 85-094 Bydgoszcz, Poland; 4Department of Laboratory Medicine, Nicolaus Copernicus University in Torun, Collegium Medicum in Bydgoszcz, 85-094 Bydgoszcz, Poland; 5Department of Family Medicine, Nicolaus Copernicus University in Torun, Collegium Medicum in Bydgoszcz, 85-094 Bydgoszcz, Poland; 6Department of Internal Medicine and Geriatric Cardiology, Center of Postgraduate Medical Education, 01-813 Warsaw, Poland; 7Department of Epidemiology and Health Promotion, School of Public Health, Center of Postgraduate Medical Education, 01-826 Warsaw, Poland

**Keywords:** total cholesterol, LDL, HDL, triglycerides, lipoprotein (a), dyslipidemia, determinants

## Abstract

To assess the determinants of lipid parameters in primary care patients without diagnosed cardiovascular disease (CVD), a cross-sectional study was conducted during 2018–2019 with a total of 200 patients. The following lipid parameters were measured: total cholesterol (TC), low-density lipoprotein cholesterol (LDL-C), high-density lipoprotein cholesterol (HDL-C), triglycerides (TG), small, dense LDL (sdLDL-C), and lipoprotein (a) (Lp(a)). Predictors of elevated and adequately controlled lipid parameters were assessed with logistic regression analysis. Older age was related to higher risk of TC ≥ 6.2 mmol/L [OR 1.03 (95% CI 1.0–1.05)], sdLDL-C ≥ 1.0 mmol/L [OR 1.05 (95% CI 1.0–1.1)], and decreased risk of Lp(a) ≥ 50 mg/dL [OR 0.97 (95% CI 0.94–0.99)]. Patients with diabetes mellitus (DM) had increased probability of TG ≥ 2.25 mmol/L [OR 3.77 (95% CI 1.34–10.6)] and Lp(a) ≥ 50 mg/dL [OR 2.97 (1.34–6.10)] as well as adequate control of TG and Lp(a). Higher material status was related to lower risk of TC ≥ 6.2 mmol/L [OR 0.19 (95% CI 0.04–0.82)] and LDL-C ≥ 3.6 mmol/L [OR 0.33 (95% CI 0.12–0.92)]. High BMI was related to increased [OR 1.14 (95% CI 1.02–1.29)], and female gender [OR 0.33 (95% CI 0.12–0.96)] and hypertension [OR 0.29 (95% CI 0.1–0.87)] to decreased risk of TG ≥ 2.25 mmol/L [OR 1.14 (95% CI 1.02–1.29)]. Taking lipid-lowering drugs (LLD) was associated with LDL-C < 2.6 mmol/L [OR 2.1 (95% CI 1.05–4.19)] and Lp(a) < 30 mg/dL [OR 0.48 (95% CI 0.25–0.93)]. Physical activity was related to LDL-C < 2.6 mmol/L [OR 2.02 (95% CI 1.02–3.98)]. Higher abdominal circumference was associated with decreased risk of TG < 1.7 mmol/L [OR 0.96 (95% CI 0.93–0.99)]. Elevated lipid parameters were related to age, gender, material status, BMI, history of DM, and hypertension. Adequate control was associated with age, education, physical activity, LLD, history of DM, and abdominal circumference.

## 1. Introduction

Cardiovascular diseases (CVD) remain one of the major causes of morbidity and mortality worldwide. The number of patients with diagnosed CVD increased from 271 million in 1990 to 523 million in 2019 [1]. To reduce the burden related to CVD multidirectional strategies should be undertaken, one of which is increased emphasis on cardiovascular (CV) prevention both at the individual and population levels [2]. Previous observational studies led to the identification of various CV risk factors, of which elevated blood pressure, elevated glycemia, smoking, low physical activity, obesity, or increased concentration of low-density lipoprotein cholesterol (LDL-C) can be modified [1,2,3]. The implementation of the guidelines on CVD prevention and achievement of therapeutic targets was assessed in recent years with five cross-sectional surveys called EUROASPIRE (European Action on Secondary and Primary Prevention by Intervention to Reduce Events) [4]. The results of the EUROASPIRE program showed unsatisfactory control of major CV risk factors, including poor control of lipid disorders. The evaluation of lipid parameters is an integral part of the assessment of the global CV risk and is included as an input in the Systemic Coronary Risk Estimation 2 (SCORE2) and SCORE2-Older Persons (SCORE2-OP) risk algorithms [2]. 

Numerous, repetitive studies confirmed that increased LDL-C is a significant CV risk factor and a reduction in LDL-C by 1 mmol/L with statins is related to a 20–25% risk reduction in myocardial infarction, stroke, coronary angioplasty, or CV death [5,6]. The abnormal concentration of other particles, e.g., increased concentration of triglycerides (TG) and lipoprotein (a) (Lp(a)) or decreased concentration of high-density lipoprotein cholesterol (HDL-C) was also related to an increased occurrence of CV events, but the evidence is less consistent [6,7,8,9,10]. Previous epidemiological studies on European and American populations showed a high prevalence of hyperlipidemia that varied between 53% and 69.2% [11,12,13,14,15]. Due to the inadequate control of hyperlipidemia shown in previous studies, a better understanding of the parameters influencing both high and low lipid concentrations is necessary [4]. The aim of this study was to assess the determinants of lipid parameters in primary care patients with CV risk factors, but without diagnosed CVD. 

## 2. Materials and Methods

Data were collected as part of the EUROASPIRE V registry, a multicenter, cross-sectional, observational study. The primary care arm in Poland was conducted at four primary care centers between spring 2018 and autumn 2019. Patients included in this study were adults, aged 18–80 years, without previously diagnosed CVD, who had been prescribed, for at least six months and not more than two years, one or more of the following treatments: antihypertensive drugs and/or lipid-lowering drugs and/or glucose-lowering treatment (diet and/or oral antidiabetic medicines and/or insulin). Patients’ eligibility for inclusion was assessed based on the medical records. 

Patients who fulfilled the inclusion criteria were invited for an interview performed by a qualified nurse or a physician. Data regarding patients’ level of education and material status were collected. Material status was subjectively described by patients as: 1—“very low”; 2—“low”; 3—“average”; 4—“high”. Level of physical activity was assessed based on patients’ answers to the following question: “Which of the following terms best describes your extra-professional activity?”, with 4 possible answers: 1—“I do not have any physical activity other than my professional work.”; 2—“Only light physical activity most of the time.”; 3—“Intensive physical activity at least 20 min 1–2 times a week.”; and 4—“20 min of vigorous physical activity more than twice a week.”. Adequate physical activity was defined as 20 min of intensive activity 1–2 times a week or more. Basic anthropometric parameters were measured: height (m), weight (kg), body mass index (BMI, kg/m^2^), and abdominal circumference (cm). Obesity was defined as BMI ≥ 30.0 kg/m^2^ and central obesity as abdominal circumference ≥ 102.0 cm for men and ≥ 88.0 cm for women. Blood pressure was measured twice on the right arm in a sitting position with a validated, automated sphygmomanometer. Elevated blood pressure was defined as systolic blood pressure (SBP) ≥ 140 mmHg and diastolic blood pressure (DBP) ≥ 90 mmHg. Smoking was defined as self-reported smoking and verified with an assessment of carbon monoxide in exhaled air. 

Fasting blood samples were collected from all study participants. The concentration of the following parameters was measured: total cholesterol (TC, mmol/L), low-density lipoprotein cholesterol (LDL-C, mmol/L), high-density lipoprotein cholesterol (HDL-C, mmol/L), triglycerides (TG, mmol/L), small, dense LDL-C (sdLDL-C, mmol/L), lipoprotein (a) (Lp(a), mg/dL), high-sensitivity cardiac troponin I (hsTnI, ng/L), C-reactive protein (CRP, mg/L), serum creatinine (mg/dL), and plasma glucose (mmol/L). Highly elevated lipid parameters were defined as TC ≥ 6.2 mmol/L (≥ 240 mg/dL), LDL-C ≥ 3.6 mmol/L (≥ 140 mg/dL), TG ≥ 2.25 mmol/L (≥ 200 mg/dL), sdLDL-C ≥ 1.0 mmol/L (≥ 40 mg/dL), and Lp(a) ≥ 50 mg/dL. Adequate control of lipids was defined as TC < 4.9 mmol/L (< 190 mg/dL), LDL-C < 2.6 mmol/L (< 100 mg/dL), TG < 1.7 mmol/L (< 150 mg/dL), sdLDL-C < 0.5 mmol/L (< 20 mg/dL), and Lp(a) < 30 mg/dL [2,6,16,17]. 

Laboratory measurements: plasma glucose, serum CRP, creatinine, and lipid profile (TC, TG, HDL-C, and LDL-C) concentrations were measured directly using an ABX Pentra 400 autoanalyzer (Horiba Medical, Montpellier, France). Concentrations of sdLDL-C and Lp(a) were assayed with the immunoturbidimetric method using Randox Lipoprotein(a) and sdLDL-Cholesterol kits (Randox Laboratories Ltd., Crumlin, UK), adapted to the ABX Pentra 400 autoanalyzer. High-sensitivity cardiac troponin I (hsTnI) assay was performed using the Alinity i platform (Abbott Laboratories, Lake Forest, IL, USA). Sex-specific cut-off values, based on 99th percentile URL, were used: 16 ng/L for females and 34 ng/L for males. Fasting glucose was measured in fluoride plasma immediately after collection. Serum samples for the remaining tests were transferred to tubes adapted for freezing and frozen at −70 °C for further analysis.

All participants signed informed consent prior to inclusion. This study received the approval of the Ethics Committee of The Nicolaus Copernicus University in Torun, Collegium Medicum in Bydgoszcz (KB 586/2017) and was conducted according to the Declaration of Helsinki and Good Clinical Practice principles. 

Statistical analysis was performed with IBM SPSS Statistic software version 27 (IBM Corp., Armonk, NY, USA). A two-sided *p*-value < 0.05 was applied for statistical significance. The Shapiro–Wilk test and the analysis of histograms were performed to determine the data distribution. Categorical variables were presented as absolute values and percentages. Continuous variables were shown as medians with interquartile range (IQR) regardless of the data distribution to increase the transparency and uniformity of the data. The differences between the two variables were analyzed using the Mann–Whitney test or Student’s *t*-test as appropriate according to data distribution; for more than two variables, the Kruskal–Wallis test or ANOVA analysis was administered. To assess the determinants of adequately controlled and highly elevated lipid parameters, a logistic regression analysis was performed. All patients were included in the analysis, regardless of the declared lipid-lowering treatment. Initially, univariate logistic regression analysis was conducted to determine the eligibility of the variables. All parameters with a *p*-value *p* < 0.1 were included in the multivariate analysis. The backward elimination method was used to create the best fit model of the multivariate regression analysis. 

## 3. Results

This study included 200 patients with a majority of women (*n* = 133, 66.5%, *p* < 0.001). The median age of the studied group was 52 years (IQR 43–60). Hypertension was the inclusion criterion in 70% of cases (*n* = 140), hyperlipidemia in 52.5% (*n* = 105), and diabetes mellitus (DM) in 20.5% (*n* = 41). The analyzed population consisted mostly of people with higher education (*n* = 117, 58.5%, *p* = 0.016) and with average material status (*n* = 141, 70.5%, *p* < 0.001). The median BMI was 26.0 kg/m^2^ (IQR 23.9–28.7 kg/m^2^) and the median abdominal circumference was 87.0 cm (IQR 70–82). Adequate physical activity was declared by 81 patients (40.5%) and the majority of the patients were non-smokers (*n* = 170, 85%). More detailed baseline information regarding the studied group is presented in Table 1.

### 3.1. Analysis of Lipid Parameters

Elevated LDL-C (> 2.6 mmol/L) was observed in 77% (*n* = 154) of cases, TG > 1.7 mmol/L in 18.5% (*n* = 37), and TC > 4.9 mmol/L was found in 76% of patients (*n* = 152). A total of 91% of patients either were previously diagnosed with hyperlipidemia or had elevated LDL-C without a previous diagnosis of lipid disorders. The studied population was characterized by elevated total cholesterol with a median of 5.56 mmol/L (IQR 4.91–6.26) and elevated LDL-C with a median of 3.29 mmol/L (IQR 2.68–4.0). The median values of the other lipid parameters are presented in Table 2. Lipid-lowering treatment was reported in 46% of cases (*n* = 92). The most used lipid-lowering agents were statins (97.8%). Rosuvastatin was applied in 50.5% and atorvastatin in 38.7% of cases. 

The analysis of the specific subgroups showed lower HDL-C (1.33 mmol/L (1.18–1.54) vs. 1.60 mmol/L (1.37–1.90), *p* < 0.001) and higher TG (1.33 mmol/L (0.96–1.75) vs. 1.13 mmol/L (0.88–1.44), *p* = 0.03) in men. Patients < 60 years of age had a lower median of TC (5.51 mmol/L (4.89–6.19) vs. 5.72 mmol/L (5.07–6.95), *p* = 0.045) in comparison to the older group. Diabetic patients had lower HDL-C (1.38 mmol/L (IQR 1.18–1.76) vs. 1.53 mmol/L (IQR 1.28–1.84), *p* = 0.036) and higher TG (1.39 mmol/L (IQR 0.97–2.14) vs. 1.14 mmol/L (IQR 0.88–1.50), *p* = 0.012). Similar observations occurred in patients with elevated BMI who had lower HDL-C and higher TG median values in comparison to the patients with an adequate BMI (Table 2). Abdominal obesity was related to a higher TG and sdLDL-C concentration. Patients with higher education had a lower TG concentration in comparison to those with a lower level of education (1.06 mmol/L (0.83–1.37) vs. 1.39 mmol/L (1.09–1.76), *p* < 0.001). A significantly lower TG concentration was observed in patients with high material status (0.96 mmol/L (0.79–1.35)) compared to the group with average and low income (1.23 mmol/L (0.92–1.56) and 1.34 mmol/L (1.03–2.08), respectively, *p* = 0.017). Patients treated with lipid-lowering drugs (LLD) had a higher Lp(a) concentration (14.04 mg/dL (4.31–50.67) vs. 6.24 mg/dL (3.12–33.21), *p* = 0.007). No other significant differences were found regarding lipid parameters (Table 2).

### 3.2. Determinants of Elevated Lipid Parameters

Results of the univariate analyses of the determinants of elevated lipid parameters are presented in the Appendix A. Figure 1 shows the results of the independent risk factors related to elevated lipid parameters based on the multivariate analysis model. Older age was associated with increased risk of TC ≥ 6.2 mmol/L [OR 1.03 (95% CI 1.0–1.05)] and sdLDL-C ≥ 1.0 mmol/L [OR 1.05 (95% CI 1.0–1.1)] but lower risk of Lp(a) > 50 mg/dL [OR 0.97 (95% CI 0.94–0.99]. Higher material status was associated with a lower risk of TC ≥ 6.2 mmol/L [OR 0.19 (95% CI 0.04–0.82)] and LDL-C ≥ 3.6 mmol/L [OR 0.33 (95% CI 0.12–0.92)]. Females [OR 0.33 (95% CI 0.12–0.96)] and patients with arterial hypertension [OR 0.29 (0.10–0.87)] were less likely to have elevated TG ≥ 2.25 mmol/L. Increased BMI [OR 1.15 (95% CI 1.01–1.31)] and DM [OR 3.77 (95% CI 1.34–10.06)] were related to increased risk of TG ≥ 2.25 mmol/L. A diagnosis of DM was an independent risk factor for Lp(a) > 50 mg/dL [OR 2.97 (95% CI 1.34–6.10)]. 

### 3.3. Determinants of Adequate Control of Lipid Parameters

Based on the univariate analysis, no significant determinant of TC < 4.9 mmol/L and sdLDL-C < 0.5 mmol/L was found. The results of the univariate analyses for determinants of LDL-C < 2.6 mmol/L, TG < 1.7 mmol/L, and Lp(a) < 30 mg/dL are presented in the Appendix A. Figure 2 shows the determinates of adequate control of lipid parameters based on the results of multivariate analyses. Patients who received lipid-lowering drugs and those who performed adequate physical activity were twice more likely to have LDL-C < 2.6 mmol/L. Higher education was associated with better control of TG [OR 2.68 (95% CI 1.23–5.86)] and higher AC was related to poor TG control [OR 0.96 (95% CI 0.93–0.99)]. Diabetic patients were less likely to have TG < 1.7 mmol/L and Lp(a) < 30 mg/dL by 60% and 63%, respectively. Taking LLD was related to a lower chance of Lp(a) < 30 mg/dL [OR 0.48 (95% CI 0.25–0.93)] and older patients were more likely to have low Lp(a) concentration [OR 1.03 (95% CI 1.00–1.06)]. 

## 4. Discussion

This study presents a complex analysis of lipid parameters in patients without diagnosed CVD but with known CV risk factors. Herein, not only the level of a major lipid fraction (TC, LDL-C, HDL-C, and TG) was analyzed but also less frequently measured sdLDL-C and Lp(a) were assessed, providing a comprehensive view of the population lipid profile. Previous observational studies reported that the prevalence of hypercholesterolemia in Poland is high and this condition might affect two-thirds of Polish adults [12,13]. Kotseva et al. showed that the prevalence of LDL-C ≥ 2.6 mmol/L was also high in other European countries and was observed in 77.5% of men and 83.3% of women not receiving lipid-lowering treatment [4]. These results are in line with those observed in the presented analysis. 

Adequate control of lipid disorders remains one of the major goals of preventive cardiology. Patients’ high adherence and functioning are crucial to achieving the therapeutic target and additional tools (e.g., the Adherence in Chronic Disease Scale or the Functioning in Chronic Illness Scale) might be helpful to identify patients at higher risk [18,19,20]. Nevertheless, long-term observation of the prevalence and management of dyslipidemias in Poland showed stagnation or worsening in the control of lipid fractions [13]. The control of hypercholesterolemia in patients with the highest CV risk is also suboptimal [21]. Intensified prevention strategies, early identification, and adequate treatment of lipid disorders are essential to improve the actual situation. A better understanding of factors related to the control of lipid parameters might be beneficial for achieving therapeutic targets. 

Age and gender are important nonmodifiable CV risk factors [2]. Results of the presented analysis showed the importance of age as a determinant of elevated TC and female gender as an independent factor reducing the risk of TG ≥ 2.25 mmol/L by 67%. Elevated median values of TC and LDL-C were observed in the older group. High levels of TC and LDL-C, particularly in middle-aged and older patients, were presented also in the NATPOL study, which investigated the prevalence and control of CVD risk factors in Poland [12]. Gender-related differences in lipid profiles are well described in the literature with higher levels of TG and lower HDL-C in men and less consistent data regarding TC and LDL-C [7,12,22,23]. Our results confirmed the higher HDL-C and lower TG in women but no differences regarding TC and LDL-C levels were found. However, there was a trend towards higher TC in women. Potential causes of differences in lipid levels between men and women may result from differences in metabolism, and hormonal balance [22]. The role of lower adherence and less aggressive treatment with lipid-lowering drugs in females could also be of importance [24,25]. The significant reduction in major vascular events due to statin-related reduction in LDL-C has been proven regardless of the baseline risk category [5]. The crucial role of lipid-lowering agents, along with performing regular physical activity, was underlined in the present study since patients receiving treatment and exercising regularly were twice more likely to have adequate control of LDL-C. Interestingly, factors related to the control of hypercholesterolemia may somewhat differ in primary compared to secondary prevention [26].

Contrary to TC and LDL-C, the control of TG in the analyzed population was much better, with only 18.5% of patients who had TG level over 1.7 mmol/L. Previous Polish studies reported hypertriglyceridemia in approximately 21% of patients [11,12]. Obesity and history of DM were independent risk factors of abnormal TG concentration in the current analysis, similar to other observational studies [7,12,23]. In the large national Korean study on patients without coronary heart disease, the group with elevated TG was older, with a higher rate of men, smokers, physically inactive, diabetic, and hypertensive individuals, as well as with higher BMI, abdominal circumference, and with elevated other lipid parameters [7]. In the Polish observational studies, TG ≥ 2.25 mmol/L was found more often in overweight and obese patients, and increased BMI was confirmed as a risk factor for hypertriglyceridemia regardless of patients’ gender [12,27]. Higher education was a determinant of adequate TG control in the current analysis. In the NATPOL study among patients with higher education, the rate of individuals with TG ≥ 2.25 mmol/L was lower [12]. On contrary, the authors of the Multi-centre National Population Health Examination Survey (WOBASZ study) found no relationship between TG concentration and level of education as well as patients’ income [27]. Interestingly, we observed that patients with hypertension had a lower risk of TG ≥ 2.25 mmol/L. This observation could be partially explained by the fact that in the group without hypertension 33% of patients had DM, which is a known risk factor for elevated TG. The prevalence of DM among patients with hypertension was two times lower (15%); therefore, the result regarding the impact of hypertension should be considered with caution. 

Although sdLDL-C and Lp(a) were related to increased CV risk, their levels are less commonly analyzed in daily clinical practice [6,17,28]. In the current analysis, the only risk factor associated with sdLDL-C ≥ 1.0 mmol/L was older age. Previous studies showed higher sdLDL-C concentration in men, Caucasians, and older patients [29,30]. Izumida et al. reported a positive, non-linear correlation between sdLDL-C and patient’s age in both men and women [31]. The highest values for men were observed in 50–54 years of age with further decrease and 60–64 years for women with a later plateau phase. The measurement of sdLDL-C requires specialized laboratories; however, the suspicion of high sdLDL-C concentration might be made based on the abnormalities in the commonly measured lipid parameters and patients at risk might be selected for further diagnostics [30]. Lp(a) is a well-known causal risk factor of atherosclerotic CVD and previous studies showed that its level is mostly related to inherited genetic factors [6]. Nevertheless, data regarding other factors are scarce. In the current analysis, younger patients and those with DM were more likely to have increased Lp(a). Furthermore, taking LLDs (97.8% were statins) was associated with a 52% lower chance of Lp(a) < 30 mg/dL. Based on recent studies, statin therapy might significantly increase the Lp(a) concentration [32]. The relationship between Lp(a) and DM remains a subject of debate. Observational and epidemiological studies suggested that low Lp(a) increases the risk of DM; however, results from Mendelian randomization studies are inconclusive [33,34]. The data on the impact of DM on Lp(a) level are also limited, in a small cross-sectional study by Ramirez et al. patients with poorly controlled DM had higher Lp(a) concentrations [35]. A modest increase in Lp(a) concentration with age, with a more pronounced rise in women over 50 years of age, was reported in a large Danish cohort [36]. Further research on non-inherited factors influencing Lp(a) is essential. 

The observational character is one of the major limitations of the presented analysis. The study design allows to investigate the association between the variables, but the inference about cause-and-effect relationships is limited. The data regarding material status and physical activity were subjectively described by the patient; therefore, there might be a risk of overestimation due to its survey-based character. Furthermore, the study group was heterogenous regarding lipid-lowering treatment that might influence the results of lipid parameters in some patients. No follow-up information on the occurrence of CV incidents is available. Nevertheless, this study provided an analysis of the burden of lipid disorders in the analyzed population of patients without CVD. 

## 5. Conclusions

In the analyzed population of patients without diagnosed CVD the control of lipid parameters should be considered unsatisfactory. A high concentration of TC and LDL-C was observed, especially in older patients. Lipid-lowering treatment, physical activity, and high material status were associated with better control of LDL-C. The determinants of highly elevated TC were older age and lower material status. Gender, DM, BMI, AC, and material status determined the control of TG. Lp(a) concentration was determined by age, history of DM, and taking lipid-lowering agents. Older patients had a higher risk of elevated sdLDL-C. This study comprehensively describes the lipid profile of the population of patients without CVD and presents the determinants of elevated lipid parameters as well as its adequate control. The knowledge regarding determinants of lipid parameters might be used in everyday clinical practice to select patients at high risk of lipid disorders and poor control and to plan specific preventive strategies. 

## Figures and Tables

**Figure 1 jcm-12-02738-f001:**
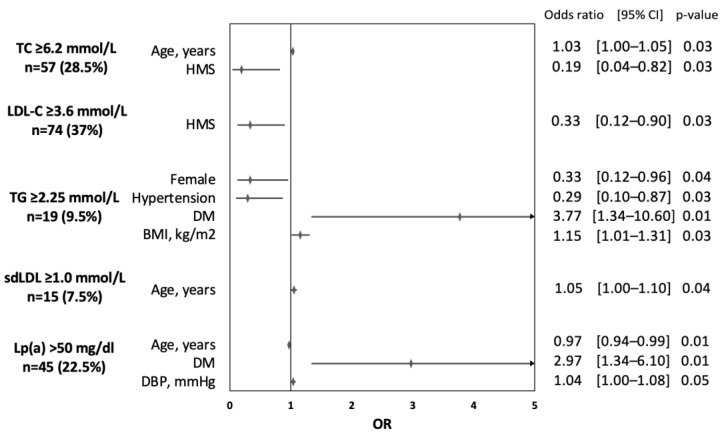
Determinants of elevated lipid parameter based on the results of multivariate analyses. Rates of patients with highly elevated lipid parameters are reported in round brackets. (HMS—higher material status; DM—diabetes mellitus; BMI—body mass index; DBP—diastolic blood pressure.)

**Figure 2 jcm-12-02738-f002:**
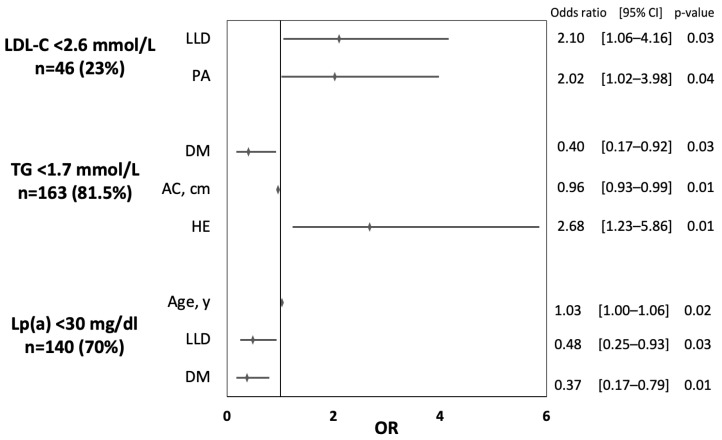
Determinants of adequate control of lipid parameter based on the results of multivariate analyses. Rates of patients with adequately controlled lipid parameters are reported in round brackets. (LLD—lipid-lowering drugs; PA—physical activity; DM—diabetes mellitus; AC—abdominal circumference; HE—higher education.)

**Table 1 jcm-12-02738-t001:** Baseline characteristic of the studied group. (AC—abdominal circumference; BMI—body mass index; BP—blood pressure; CRP—C-reactive protein; DBP—diastolic blood pressure; DM—diabetes mellitus; F—females; FPG—fasting plasma glucose; hsTnI—high-sensitivity cardiac troponin I; IQR—interquartile range; M—males; SBP—systolic blood pressure.)

Analyzed Parameter		Value
Age, median (IQR)		52.0 (43.0–60.0)
Years of education, median (IQR)		15.0 (13.0–17.0)
Higher education, *n* (%)		117 (58.5)
Female gender, *n* (%)		133 (66.5)
Material status, *n* (%)	Low	30 (15.0)
	Average	141(70.5)
	High	29 (14.5)
History of hyperlipidemia, *n* (%)		105 (52.5)
Lipid-lowering drugs, *n* (%)		92 (46.0)
History of hypertension, *n* (%)		140 (70.0)
BP-lowering drugs, *n* (%)		126 (63.0)
SBP [mmHg], median (IQR)		125.0 (118.0–135.0)
DBP [mmHg], median (IQR)		77.0 (70.0–82.0)
SBP/DBP < 140/90 mmHg, *n* (%)		155 (77.5)
History of DM, *n* (%)		41 (20.5)
Insulin and/or oral medications, *n* (%)		17 (8.5)
FPG [mmol/L], median (IQR)		5.42 (5.04–5.88)
FPG < 5.56 mmol/L (<100 mg/dL), *n* (%)		117 (58.5)
AC [cm], median (IQR)		87.0 (80.0–95.5)
AC, *n* (%)	Normal	74 (37.0)
	Overweight [F ≥ 80 cm and <88 cm, M ≥ 94 cm and <102 cm]	57 (28.5)
	Obesity [F ≥ 88 cm, M ≥ 102 cm]	69 (34.5)
BMI [kg/m^2^], median (IQR)		26.0 (24.0–28.7)
BMI category, *n* (%)	Underweight < 18.5 kg/m^2^	6 (3.0)
	Normal 18.5–24.9 kg/m^2^	77 (38.5)
	Overweight 25.0–29.9 kg/m^2^	84 (42.0)
	Obesity > 30.0 kg/m^2^	33 (16.5)
Non-smoker status, *n* (%)		170 (85.0)
Adequate physical activity, *n* (%)		81 (40.5)
hsTnI [ng/L], median (IQR)		2.3 (1.6–3.2)
CRP [mg/L], median (IQR)		0.9 (0.6–1.7)
Serum creatinine [mg/dL], median (IQR)		0.8 (0.7–0.9)

**Table 2 jcm-12-02738-t002:** Median and interquartile range of analyzed lipid parameters in the general population and in the subgroups. (AC—abdominal circumference; BMI—body mass index; DM—diabetes mellitus; HDL-C—high-density lipoprotein cholesterol; HT—arterial hypertension; LDL-C—low-density lipoprotein cholesterol; Lp(a)—lipoprotein (a); sdLDL-C—small, dense LDL-C; TC—total cholesterol; TG—triglycerides.)

Subgroups		TC [mmol/L]	LDL-C [mmol/L]	HDL-C [mmol/L]	TG [mmol/L]	sdLDL-C [mmol/L]	Lp(a) [mg/dL]
Total population		5.56 (4.91–6.26)	3.29 (2.68–4.0)	1.50 (1.25–1.81)	1.21 (0.90–1.55)	0.64 (0.53–0.78)	9.19 (3.54–42.07)
Gender	Female	5.64 (5.02–6.43)	3.30 (2.69–4.17)	1.60 (1.37–1.90)	1.13 (0.88–1.44)	0.63 (0.52–0.79)	8.20 (3.45–40.6)
	Male	5.45 (4.77–6.08)	3.13 (2.6–3.89)	1.33 (1.18–1.54)	1.33 (0.96–1.75)	0.65 (0.53–0.79)	11.12 (3.77–42.2)
	*p*-value	0.055	0.19	< 0.001	0.03	0.96	0.62
Age	< 60 years	5.51 (4.89–6.19)	3.15 (2.66–3.87)	1.47 (1.27–1.78)	1.15 (0.88–1.51)	0.63 (0.53–0.75)	10.55 (3.48–61.01)
	≥ 60 years	5.72 (5.07–6.95)	3.45 (2.72–4.51)	1.59 (1.26–1.87)	1.30 (0.96–1.64)	0.66 (0.53–0.90)	6.60 (3.64–14.21)
	*p*-value	0.045	0.057	0.287	0.125	0.084	0.171
DM	No	5.55 (4.94–6.23)	3.30 (2.70–4.03)	1.53 (1.28–1.84)	1.14 (0.88–1.50)	0.63 (0.52–0.77)	8.30 (3.48–35.59)
	Yes	5.57 (5.07–6.42)	3.28 (2.58–3.83)	1.38 (1.18–1.76)	1.39 (0.97–2.14)	0.66 (0.54–0.80)	13.30 (3.98–78.80)
	*p*-value	0.906	0.721	0.036	0.012	0.429	0.094
HT	No	5.65 (5.15–6.61)	3.29 (2.69–4.17)	1.54 (1.23–1.77)	1.30 (0.97–1.78)	0.69 (0.54–0.86)	14.21 (4.05–52.86)
	Yes	5.51 (4.82–6.21)	3.29 (2.68–3.92)	1.48 (1.27–1.83)	1.15 (0.88–1.51)	0.60 (0.52–0.75)	6.81 (3.46–37.4)
	*p*-value	0.080	0.596	0.670	0.105	0.061	0.078
BMI categories	Normal	5.62 (5.0–6.27)	3.3 (2.71–3.93)	1.59 (1.35–1.89)	1.09 (0.85–1.35)	0.61 (0.51–0.73)	11.12 (3.74–64.58)
	Overweight	5.54 (4.85–6.34)	3.28 (2.66–4.08)	1.44 (1.24–1.76)	1.3 (0.93–1.61)	0.66 (0.55–0.81)	10.55 (3.61–33.33)
	Obesity	5.4 (4.75–6.18)	3.17 (2.62–3.93)	1.4 (1.19–1.65)	1.35 (0.96–2.15)	0.64 (0.53–0.81)	5.68 (3.0–23.48)
	*p*-value	0.409	0.850	0.043	0.012	0.173	0.162
AC categories	Normal	5.46 (4.65–6.12)	3.14 (2.61–3.7)	1.5 (1.24–1.81)	1.07 (0.88–1.36)	0.58 (0.51–0.72)	11.96 (4.06–64.0)
	Overweight	5.68 (5.11–6.27)	3.42 (2.82–4.07)	1.59 (1.36–1.81)	1.24 (0.89–1.52)	0.66 (0.53–0.77)	10.4 (3.57–49.6)
	Obesity	5.59 (4.87–6.52)	3.17 (2.6–4.24)	1.44 (1.24–1.89)	1.35 (0.96–1.8)	0.66 (0.54–0. 091)	6.9 (3.0–24.37)
	*p*-value	0.258	0.259	0.418	0.023	0.038	0.228
Education	Primary/ Secondary	5.54 (5.02–6.27)	3.27 (2.71–4.16)	1.44 (1.24–1.71)	1.39 (1.09–1.76)	0.66 (0.53–0.81)	8.36 (3.0–40.6)
	Higher	5.58 (4.88–6.22)	3.29 (2.63–3.89)	1.52 (1.29–1.89)	1.06 (0.83–1.37)	0.60 (0.52–0.76)	10.02 (3.74–41.6)
	*p*-value	0.810	0.332	0.123	< 0.001	0.241	0.393
Material status	Very low/Low	5.58 (5.07–6.68)	3.38 (2.72–4.19)	1.43 (1.21–1.88)	1.34 (1.03–2.08)	0.67 (0.56–0.87)	5.82 (3.22–51.72)
	Average	5.69 (4.97–6.36)	3.35 (2.63–4.06)	1.55 (1.29–1.81)	1.23 (0.92–1.56)	0.64 (0.52–0.78)	10.02 (3.6–42.25)
	High	5.35 (4.8–5.61)	3.09 (2.74–3.41)	1.42 (1.27–1.58)	0.96 (0.79–1.35)	0.57 (0.53–0.73)	11.12 (3.86–24.42)
	*p*-value	0.078	0.239	0.302	0.017	0.407	0.846
Physical activity	No	5.68 (5.09–6.31)	3.40 (2.79–4.04)	1.53 (1.27–1.87)	1.24 (0.93–1.56)	0.65 (0.53–0.80)	8.42 (3.55–44.70)
	Yes	5.42 (4.68–6.22)	3.09 (2.46–3.94)	1.43 (1.23–1.76)	1.10 (0.88–1.51)	0.58 (0.53–0.76)	9.57 (3.66–39.13)
	*p*-value	0.165	0.120	0.082	0.374	0.242	0.995
Smoking status	Non-smoker	5.51 (4.88–6.26)	3.16 (2.65–3.93)	1.50 (1.26–1.82)	1.20 (0.87–1.53)	0.63 (0.52–0.78)	9.80 (3.60–41.60)
	Active smoker	6.03 (5.28–6.22)	3.50 (2.75–4.19)	1.51 (1.24–1.81)	1.28 (1.04–1.69)	0.67 (0.58–0.78)	7.03 (3.26–47.15)
	*p*-value	0.151	0.234	0.855	0.083	0.256	0.825
Lipid-lowering treatment	No	5.57 (4.98–6.22)	3.30 (2.75–3.90)	1.50 (1.31–1.83)	1.12 (0.85–1.51)	0.63 (0.51–0.75)	6.24 (3.12–33.21)
	Yes	5.54 (4.84–6.41)	3.14 (2.38–4.19)	1.46 (1.21–1.77)	1.30 (0.96–1.60)	0.66 (0.53–0.81)	14.04 (4.31–50.67)
	*p*-value	0.758	0.613	0.162	0.073	0.247	0.007

## Data Availability

Data are available upon reasonable request. All data relevant to this study are included in the article. The original data are available from the corresponding author, within the limits of the signed informed consent from the contributors.

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
