# Peer review of "Determinants of Lipid Parameters in Patients without Diagnosed Cardiovascular Disease—Results of the Polish Arm of the EUROASPIRE V Survey"

_jcm, 2023, doi:10.3390/jcm12072738_

Round 1

Reviewer 1 Report

In the manuscript (MS) entitled “Determinants of Lipid Parameters in Patients without Diagnosed Cardiovascular Disease—Results of the Polish Arm of the EUROASPIRE V Survey”, Jakub Ratajczak and colleagues present cross-sectional study on predictive factors of abnormal vs normal lipid parameters in adult outpatient population without diagnosed cardiovascular disease (CVD). The study was conducted in the Polish cohort of primary care patients included in the ongoing EUROASPIRE V Survey established by the European Society of Cardiology.

The key finding of the study is that in this population of patients without diagnosed CVD elevated lipid parameters are associated with age, sex, socioeconomic status, BMI, history of DM and hypertension, with older age appearing to be the most significant factor.

In general, the MS (including the supplementary material) is concise and well written. My main concerns are related to insufficient details on biochemistry measurements and description of some of the results. Please, find my questions/suggestions below.

Major points:

1. Please, provide more details on lipid measurements - what type of essays/equipment was used? Was LDL-C calculated (based on what formula?) or directly measured?

2. Please, provide information on troponin assay and cut-off values for the assay used in this study population.

3. Were blood sampled stored for further analysis in one core lab or were they immediately analysed in each centre participating in the study?4

4. Clarify if hypertension (HT) in the analysis of the results is defined as elevated BP > 140/90 mm Hg or if this also includes hypertensive patients on pharmacotherapy with normal BP.

5. Clarify what is the cut-off value for older age. Was older age prespecified before the analysis of results or was it an outcome of analysis?

6. Conclusions appear to be a condensed description of main results only. I suggest including some highlight of the novelty/importance of this analysis for the field of CVD and lipid disturbances.

7. Please, indicate in the MS limitations of the use of the survey-based evaluation of socio-economic status and physical activity, as these are often overestimated by participants. 

 Minor:

“All participants have signed informed consent prior to the inclusion” probably should be “All participants signed informed consent prior to the inclusion”, otherwise it suggests that the process of providing informed consent is not finished (line 115).

Author Response

Dear Editors and Reviewers,

We would like to thank you for your thoughtful and helpful review of the manuscript: “Determinants of Lipid Parameters in Patients without Diagnosed Cardiovascular Disease—Results of the Polish Arm of the EUROASPIRE V Survey”. We strongly appreciate your interest in our paper. We answered all questions and responded to all issues raised by the Reviewers. As requested, the changes to the text are tracked in the revised manuscript, we also outlined them in red italics. We believe that after corrections suggested by the reviewers, the quality of our paper has improved. We hope that in its current form you will find it suitable for publication in your journal.

We look forward to your replay at your earliest convenience.

Sincerely,

Jakub Ratajczak

Jakub Ratajczak, MD

Department of Cardiology and Internal Medicine,

Department of Cardiac Rehabilitation Health Promotion,

Nicolaus Copernicus University in Torun, Collegium Medicum in Bydgoszcz,

  1. Marii Skłodowskiej-Curie 9, 85-094 Bydgoszcz, Poland,

e-mail: [email protected] ; tel.: +48 52 585 40 23; fax: +48 52 585 40 24

 Response to Reviewer 1 Comments:

In the manuscript (MS) entitled “Determinants of Lipid Parameters in Patients without Diagnosed Cardiovascular Disease—Results of the Polish Arm of the EUROASPIRE V Survey”, Jakub Ratajczak and colleagues present cross-sectional study on predictive factors of abnormal vs normal lipid parameters in adult outpatient population without diagnosed cardiovascular disease (CVD). The study was conducted in the Polish cohort of primary care patients included in the ongoing EUROASPIRE V Survey established by the European Society of Cardiology.

The key finding of the study is that in this population of patients without diagnosed CVD elevated lipid parameters are associated with age, sex, socioeconomic status, BMI, history of DM and hypertension, with older age appearing to be the most significant factor.

In general, the MS (including the supplementary material) is concise and well written. My main concerns are related to insufficient details on biochemistry measurements and description of some of the results. Please, find my questions/suggestions below.

Major points:

  1. Please, provide more details on lipid measurements - what type of essays/equipment was used? Was LDL-C calculated (based on what formula?) or directly measured?

Response: Laboratory measurements: plasma glucose, serum CRP, creatinine, and lipid profile (TC, TG, HDL-C and LDL-C) concentrations were measured directly using ABX Pentra 400 autoanalyzer (Horiba Medical, Montpellier, France). Concentration of sdLDL-C and Lp(a) were assayed with immunoturbidimetric method using Randox Lipoprotein(a) and sdLDL-Cholesterol kits (Randox Laboratories Ltd., Crumlin, UK), adapted to the ABX Pentra 400 autoanalyzer. Details regarding laboratory measurements were added to the “Materials and Methods” section of the main manuscript.

  1. Please, provide information on troponin assay and cut-off values for the assay used in this study population.

Response: High sensitivity cardiac troponin I (hsTnI) assay was performed using Alinity i platform (Abbott Laboratories, lake Forest, IL, USA). Sex-specific cut-off values, based on 99th percentile URL, were used: 16 ng/L for females and 34 ng/L for males. Details regarding laboratory measurements were added to the “Materials and Methods” section of the main manuscript.

  1. Were blood sampled stored for further analysis in one core lab or were they immediately analysed in each centre participating in the study?

Response: Fasting glucose was measured in fluoride plasma immediately after collection. Serum samples for the remaining tests were transferred to tubes adapted for freezing and frozen at -70°C for further analysis. Details regarding laboratory measurements were added to the “Materials and Methods” section of the main manuscript.

  1. Clarify if hypertension (HT) in the analysis of the results is defined as elevated BP > 140/90 mm Hg or if this also includes hypertensive patients on pharmacotherapy with normal BP.

Response: Hypertension in the analysis was defined as the diagnosis and treatment of hypertension regardless of the BP measured during the study. This parameter is reflected as “History of hypertension” in Table 1.

  1. Clarify what is the cut-off value for older age. Was older age prespecified before the analysis of results or was it an outcome of analysis?

Response: We assume that the reviewer refers to the “older age” term used in the section describing the analysis of the determinants of lipid parameters by logistic regression. In this case, age was a continuous variable; therefore, a specific cut-off value was not established. In each case, where “age” was a significant determinant of lipid parameters, the risk increases or decreases with each year that the patients get older. However, we also compared the lipid profile in various subgroups (Table 2.), where differences regarding age groups were analyzed. Herein, we prespecified before the analysis two age groups (<60 years and ≥60 years).

  1. Conclusions appear to be a condensed description of main results only. I suggest including some highlight of the novelty/importance of this analysis for the field of CVD and lipid disturbances.

Response: Thank you for the review. As suggested (also by the Reviewer 2) we added a sentence that underline the importance of the results in the “Conclusions” section.

  1. Please, indicate in the MS limitations of the use of the survey-based evaluation of socio-economic status and physical activity, as these are often overestimated by participants.

Response: Thank you for this remark. As suggested by the reviewer we added the information about of self-based evaluation of some parameters in the limitations section.

Minor:

“All participants have signed informed consent prior to the inclusion” probably should be “All participants signed informed consent prior to the inclusion”, otherwise it suggests that the process of providing informed consent is not finished (line 115).

Response: We have changed the sentence according to the reviewer’s suggestion.

Reviewer 2 Report

Observational study with data from a registry. The findings are interesting.

However I would like the authors highlight why they are novel and how they can be used for future studied or in clinical practice

Author Response

Dear Editors and Reviewers,

We would like to thank you for your thoughtful and helpful review of the manuscript: “Determinants of Lipid Parameters in Patients without Diagnosed Cardiovascular Disease—Results of the Polish Arm of the EUROASPIRE V Survey”. We strongly appreciate your interest in our paper. We answered all questions and responded to all issues raised by the Reviewers. As requested, the changes to the text are tracked in the revised manuscript, we also outlined them in red italics. We believe that after corrections suggested by the reviewers, the quality of our paper has improved. We hope that in its current form you will find it suitable for publication in your journal.

We look forward to your replay at your earliest convenience.

Sincerely,

Jakub Ratajczak

Jakub Ratajczak, MD

Department of Cardiology and Internal Medicine,

Department of Cardiac Rehabilitation Health Promotion,

Nicolaus Copernicus University in Torun, Collegium Medicum in Bydgoszcz,

  1. Marii Skłodowskiej-Curie 9, 85-094 Bydgoszcz, Poland,

e-mail: [email protected] ; tel.: +48 52 585 40 23; fax: +48 52 585 40 24

 Response to Reviewer 2 Comments:

Observational study with data from a registry. The findings are interesting.

However I would like the authors highlight why they are novel and how they can be used for future studied or in clinical practice

Response: Thank you for the review. As suggested (also by the Reviewer 1) we added a sentence that underline the importance of the results in the “Conclusions” section.

Reviewer 3 Report

Dear editor, dear authors,

Ratajczak's work provides important specific information on Poland and confirms the international state of knowledge.

Only minimal proposals for changes would be to relate the low smoking rate in work to the normal population in the limitations. Furthermore, as a very modifiable risk factor, the topic should first be mentioned more clearly. The method and result parts are otherwise clear and easy to understand.

Author Response

Dear Editors and Reviewers,

We would like to thank you for your thoughtful and helpful review of the manuscript: “Determinants of Lipid Parameters in Patients without Diagnosed Cardiovascular Disease—Results of the Polish Arm of the EUROASPIRE V Survey”. We strongly appreciate your interest in our paper. We answered all questions and responded to all issues raised by the Reviewers. As requested, the changes to the text are tracked in the revised manuscript, we also outlined them in red italics. We believe that after corrections suggested by the reviewers, the quality of our paper has improved. We hope that in its current form you will find it suitable for publication in your journal.

We look forward to your replay at your earliest convenience.

Sincerely,

Jakub Ratajczak

Jakub Ratajczak, MD

Department of Cardiology and Internal Medicine,

Department of Cardiac Rehabilitation Health Promotion,

Nicolaus Copernicus University in Torun, Collegium Medicum in Bydgoszcz,

  1. Marii Skłodowskiej-Curie 9, 85-094 Bydgoszcz, Poland,

e-mail: [email protected] ; tel.: +48 52 585 40 23; fax: +48 52 585 40 24

  Response to Reviewer3 Comments:

Dear editor, dear authors,

Ratajczak's work provides important specific information on Poland and confirms the international state of knowledge.

Only minimal proposals for changes would be to relate the low smoking rate in work to the normal population in the limitations. Furthermore, as a very modifiable risk factor, the topic should first be mentioned more clearly. The method and result parts are otherwise clear and easy to understand.

Response: Thank you very much for the review. However, we would like to disagree that the low smoking rate found in our study should be considered a limitation. We agree that previous reports showed a slightly higher prevalence of smoking in Poland – e.g. Pinkas et a. [Int J Environ Res Public Health. 2019 Nov 30;16(23):4820] reported daily tobacco smoking in 21% of participants. In our study, active smoking was reported in 15% of cases, but, in our opinion, this low rate should not be considered a limitation but more as a characteristic feature of the studied group.